# Spatially controllable and mechanically switchable isomorphous organoferroeleastic crystal optical waveguides and networks

Subham Ranjan [1], Avulu Vinod Kumar[2], Rajadurai Chandrasekar [2] ✉ & Satoshi Takamizawa [1] ✉

The precise, reversible, and diffusionless shape-switching ability of organic ferroelastic crystals, while maintaining their structural integrity, positions them as promising materials for next-generation hybrid photonic devices. Herein, we present versatile bi-directional ferroelasticity and optical waveguide properties of three isomorphous, halogen-based, Schiff base organic crystals. These crystals exhibit sharp bending at multiple interfaces driven by molecular movement around the CH = N bond and subsequent 180° rotational twinning, offering controlled light path manipulation. The ferroelastic nature of these crystals allowed the construction of robust hybrid photonic structures, including Z-shaped configurations, closed-loop networks, and staircase-like hybrid optical waveguides. This study highlights the potential of shape-switchable organoferroelastic crystals as waveguides for applications in programmable photonic devices.

Ferroelasticity, akin to ferroelectricity and ferromagnetism, involves spontaneous strain generation in twinned domains under external stress[1,2]. In materials science, the controllable twin domain structure in ferroelasticity finds applications in memory storage, thermal management, optoelectronic devices, and more[3–8]. The emergence of ferroelasticity in organic crystals has attracted interest due to their ease of preparation, mechanical flexibility, lightweight, high refractive index, and environmental advantages compared to the inorganic materials[9]. Several organic ferroelastic crystals have been studied with systematic structural-property analysis[10–14]. Organic crystals can undergo shape transformation through various mechanical deformations, like elastic, plastic, and helical twisting, influenced by structural factors *viz* slip planes, defects, and non-covalent interactions[15–19]. In contrast, organic ferroelastic crystals exhibit twinning-based, diffusionless phase transformations, preserving structural integrity[9,20]. The reversibility and diffusionless characteristics of organic ferroelastic crystals position them as promising materials for the futuristic structurally dynamic flexible organic crystals (FOCs).

The growing demand for miniaturized optical devices, such as optical links and wearable sensors, necessitates the development of flexible micron-sized optical waveguides (OWGs). FOCs have been extensively studied for constructing OWGs, which are essential in photonic integrated circuits[21–23]. Thiacyanine dye nanofibers exhibit potential as flexible active OWGs[24]. Recently, FOCs have demonstrated passive OWG properties similar to silicon waveguides[25,26]. The optical waveguiding in millimeter-sized elastic crystals has also illustrated the potential of FOCs as photonic device components[27], soft robots[28], and sensors[29]. Further, epitaxially grown organic heterostructures serve as multi-color emissive OWGs[30,31]. Recent advancements, such as pseudo-plasticity and mechanical micromanipulation techniques, have allowed the construction of various optical modules and photonic integrated circuits using FOCs[32–35]. However, developing flexible organic OWGs with sharp bends, preserving crystalline integrity in deformed areas, and reversibility remains challenging with existing elastic/plastic crystals. Our previous research has shown that versatile ferroelastic deformation in organic crystals, achieved through multiple mechanical modes, enables the

[1]Department of Materials System Science, Graduate School of Nanobioscience, Yokohama City University, 22-2 Seto, Kanazawa-ku, Yokohama, Kanagawa 236-0027, Japan. [2]School of Chemistry and Centre for Nanotechnology, University of Hyderabad, Prof. C. R. Rao Road, Gachibowli, Hyderabad 500 046 Telangana, India. ✉e-mail: r.chandrasekar@uohyd.ac.in; staka@yokohama-cu.ac.jp

precise creation of sharp bends at multiple interfaces[13]. Particularly, the use of photoluminescent ferroelastic crystals as OWGs represents a significant advancement[10], merging ferroelasticity with photonics for innovative hybrid optics device applications. High refractive index ferroelastic organic crystals, with light transduction control, satisfy the requisites for reversibility, crystalline integrity, and precise shape reconfiguration, enabling the advancement of spatially reconfigurable smart devices and circuits.

Recent studies have explored the use of magnetic field[36] and AFM-tip assistance[32] for achieving directional reconfigurability in OWs based on FOCs. However, a new frontier lies in leveraging ferroelastic crystals to create spatially controllable photonic structures with predictable bending angles, yet programmable through reversible phase transformation. In this work, we synthesized three isomorphous Schiff bases, each containing 3-bromo-2-{(E)-[(4-halo-phenyl)imino]methyl}−6-methoxyphenol with fluoro, chloro, and bromo substituents (1-X; where X = F, Cl, and Br). In the crystalline state, these compounds not only demonstrated shear-induced bi-directional ferroelastic deformation but also displayed a rarely observed photoluminescence (PL). The bi-directional ferroelasticity and optical characteristics enable the development of the organoferroelastic crystal OWGs with meticulous mechanical control over their sharp bends. The terminal-to-terminal alignment of three ferroelastic crystals with decreasing widths facilitates light transmission through progressively narrowing cross-sections. Furthermore, the ability to reversibly deform these OWGs and construct open and closed-loop networks suggests their potential applicability in advanced mechanophotonic devices and mechanically responsive optical circuits.

## Results and discussion

The Schiff bases (1-X) were synthesized by dissolving equimolar amounts of 6-bromo-2-hydroxy-3-methoxybenzaldehyde with their respective anilines (4-fluoroaniline, 4-chloroaniline, and 4-bromoaniline) through a nucleophilic addition reaction in hot acetonitrile (Fig. 1a). In all cases, the resulting solution was allowed to slowly evaporate, yielding millimeter-sized block-shaped, orange-colored single crystals of 2–5 mm in length, 20–60 μm in thickness, and 30–120 μm in width. Subsequent single-crystal X-ray analysis (SXRD) confirmed the identity of these Schiff bases, which were found to crystallize in a monoclinic crystal system with the $P2_1/n$ space group with isomorphous nature at room temperature (Supplementary Figs. 1–4 and Supplementary Tables 1, 2).

Investigation of the mechanical properties of the single crystals of 1-Cl showed that upon application of stress on the (00$\bar{1}$) plane, it exhibited a versatile bi-directional ferroelastic deformation (Supplementary Fig. 5, and Supplementary Movie 1 and 2). As a result, the crystals of 1-Cl could be deformed at various points, resulting in the formation of a distinct staircase pattern (Fig. 1b, c). On passing light in a 1-Cl deformed crystal, the light propagated from one end to another through the mother and daughter domains, reflecting the light-guiding ability in the sharply bent ferroelastic crystal (Supplementary Fig. 6, Supplementary Movie 3). Moreover, surface roughness measurements by 3D laser confocal microscopy revealed that the mother domain remained unaltered at the microscopic level even after undergoing ferroelastic deformation, indicating surface integrity (Fig. 1d and Supplementary Fig. 7 and 8). The surface roughness of unevenly deformed 1-Cl crystal was found in the range of 13 − 49 nm, obtained

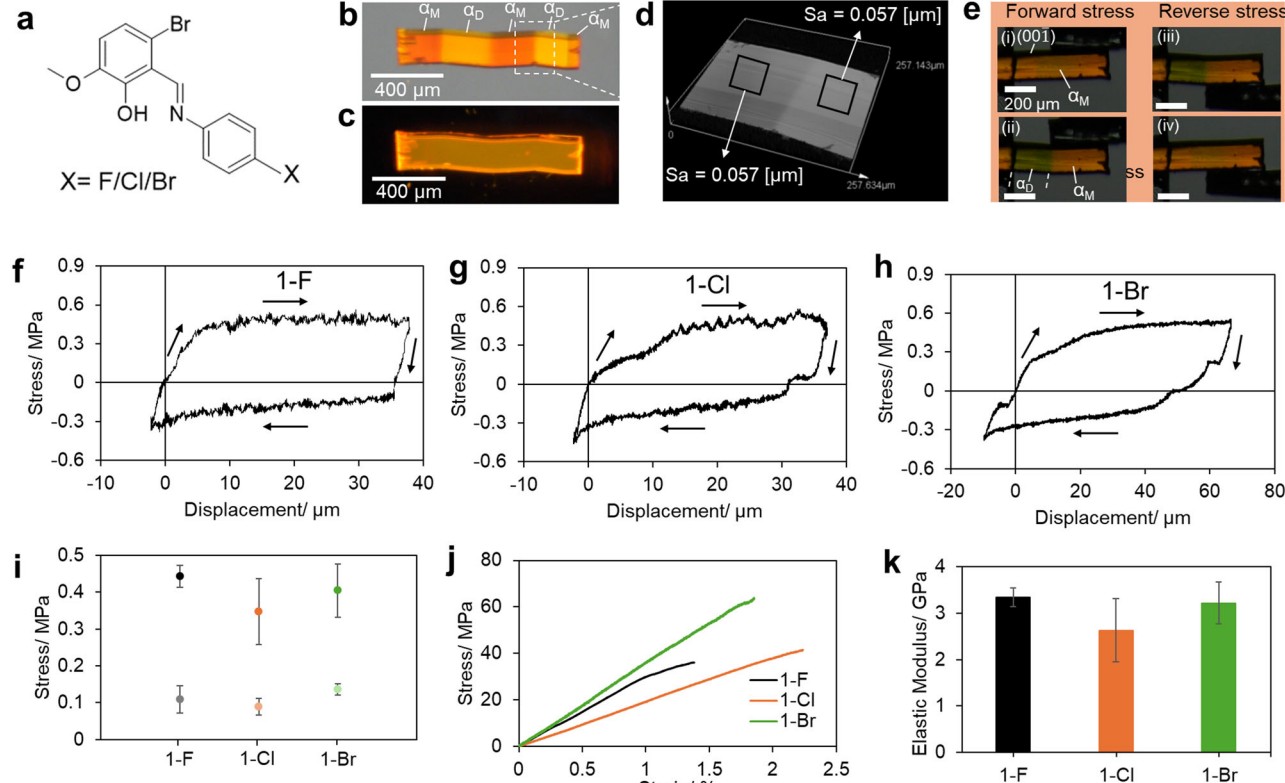

**Fig. 1 | Molecular structures and mechanical properties of 1-X (X = F, Cl, and Br).** **a** Molecular structure of Schiff bases (1-F, 1-Cl, and 1-Br). **b, c** Ferroelastically deformed crystal of 1-Cl under polarized white light and UV light, respectively. α$_M$ and α$_D$ correspond to mother and daughter domains in ferroelastically deformed crystals, respectively. **d** Surface roughness of mother and daughter domain of ferroelastically deformed crystal of 1-Cl. **e** Snapshots of 1-Cl crystal during the stress-displacement measurements. **f**–**h** Stress-displacement curves of a single crystal of 1-F, 1-Cl, and 1-Br, respectively. **i** Coercive and reverse stress of ferroelastic deformation of 1-F, 1-Cl, and 1-Br, and (**j, k**) elastic moduli of 1-F, 1-Cl, and 1-Br. The forward stress, reverse stress, and elastic modulus values are expressed as mean ± standard deviation, n = 3.

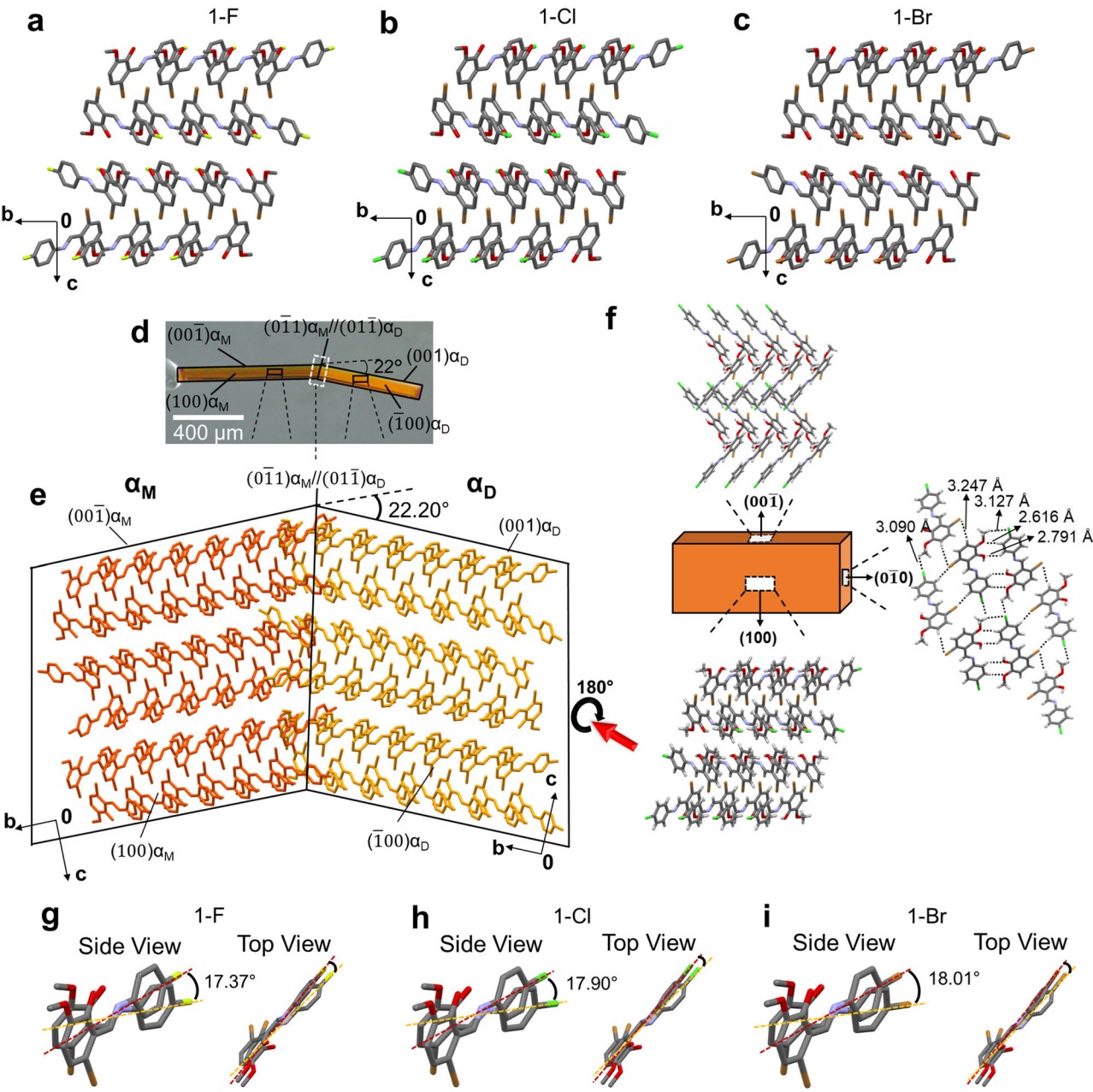

**Fig. 2 | Structural basis for the ferroelastic deformation in 1-X (X = F, Cl, Br) crystals. a–c** Crystal packing of **1-F**, **1-Cl**, and **1-Br**. **d** Face indexes of deformed single crystal **1-Cl**. **e** Estimated connecting manners of an $\alpha_M$ (mother domain) and $\alpha_D$ (daughter domain) of **1-Cl** based on the X-ray diffraction measurements. **f** Crystal packing of **1-Cl**, and (**g–i**) Estimated molecular movements at the interface of **1-F**, **1-Cl**, and **1-Br**.

by atomic force microscopy (Supplementary Fig. 23). The ferroelastic deformation of the crystals was quantitatively confirmed by applying shear stress on the (00$\bar{1}$) plane of **1-F**, **1-Cl**, and **1-Br** crystals (Fig. 1e, Supplementary Movies 4 and 5). The nucleation and propagation of the daughter domain ($\alpha_D$) from the mother domain ($\alpha_M$) were initiated at the coercive stresses of 0.44, 0.35, and 0.40 MPa, respectively (Fig. 1f–i). The reverse transformation of the daughter domain to the mother domain was initiated by applying shear stress of 0.11, 0.09, and 0.135 MPa on the (001) plane for **1-F**, **1-Cl**, and **1-Br**, respectively. Ferroelastic deformation enables strain release and reversibility. In all cases, shear-displacement graphs showed hysteresis loops, confirming ferroelastic deformation (Supplementary Fig. 9). Moreover, the coercive and reverse stresses were notably similar in each crystal, with **1-F** having the highest values, followed by **1-Br**, and **1-Cl** (Fig. 1i).

Furthermore, the three-point bending test determined the elastic moduli of these crystals, with **1-F** having the highest modulus (3.35 GPa), followed by **1-Br** (3.22 GPa), and **1-Cl** (2.63 GPa) (Fig. 1j, k, Supplementary Fig. 10, and Supplementary Movies 6 and 7). The similarity in their mechanical properties can be attributed to their isomorphous nature. Additionally, all three ferroelastic crystals exhibited rare PL under UV light. Notably, both the ferroelastic deformation quantified by the stress-strain curve and elastic moduli remained consistent when measured under UV light, indicating the crystals' photochemical stability (Supplementary Fig. 11, and Supplementary Movies 5–7).

To understand the underlying mechanism of the three isomorphous ferroelastic crystals (**1-F**, **1-Cl**, and **1-Br**), SXRD analyses were conducted (Supplementary Fig. 1, and Supplementary Table 1). All

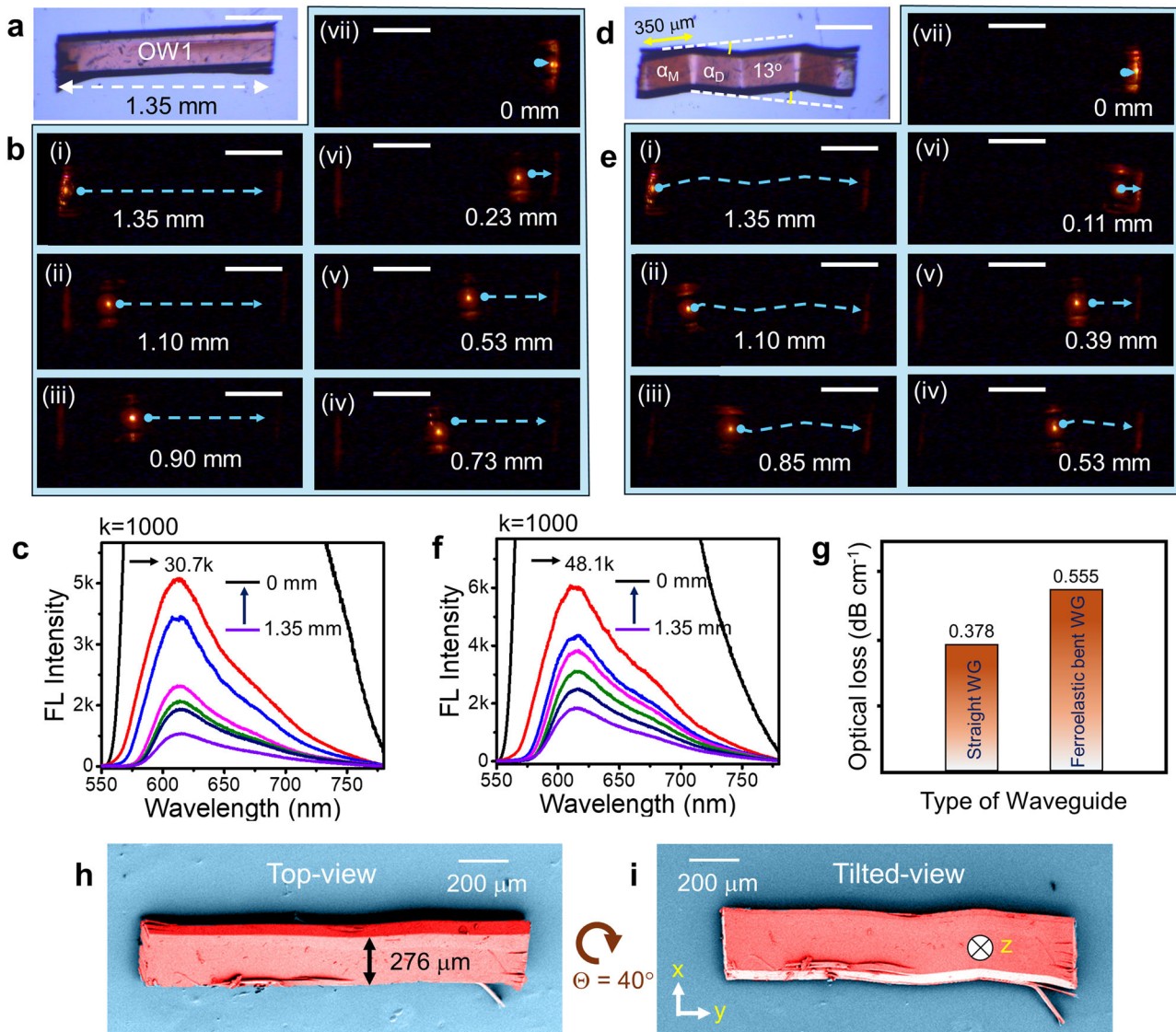

**Fig. 3 | Optical waveguiding characteristics of straight and ferroelastic bent 1-Cl crystal waveguide. a, b** Confocal optical and FL microscope images (upon excitation with 405 nm laser) of a straight **1-Cl** crystal optical waveguide (OW1), respectively. Scale bars are 360 μm. **d, e** Confocal optical and FL microscope images of a ferroelastic bent **1-Cl** OW1, respectively. Scale bars are 360 μm. **c, f** FL spectra obtained as a function of the distance between illumination and collection point in the straight and ferroelastic bent OW1. **g** Optical loss measured for straight and ferroelastic bent OW1. **h, i** Top and tilted-view color-coded FESEM images of a bent **1-Cl** OW1, respectively. Θ represents the change in the viewing angle between the top- and the tilted-views.

three crystals are referred to as the mother domains ($\alpha_M$) in their initial state. Despite variations in the halogen group (F, Cl, and Br), the crystal packing of **1-F**, **1-Cl**, and **1-Br** is similar (Fig. 2a–c). The resemblances in crystal packing, Xpac, and unit cell similarity index indicate their isomorphous nature (Fig. 2a–c, Supplementary Figs. 2–4, and Supplementary Table 2). The crystal lattice of **1-F**, **1-Cl**, and **1-Br** is predominantly stabilized through C–H⋯O hydrogen bonds, C–H⋯F/Cl/Br weak hydrogen bonds, and π⋯π stacking interactions.

The bent crystals of **1-F**, **1-Cl**, and **1-Br** revealed bi-directional ferroelastic deformation attributed to 180° rotational mechanical twinning, facilitated by molecular movement around the CH = N bond at angles of 17.37°, 17.90°, and 18.01°, respectively (Fig. 2g–i). A bending angle of 22.20° between $\alpha_M$ and $\alpha_D$ can be anticipated based on crystallographic findings at the twin interfaces $(0\bar{1}1)\,\alpha_M//(01\bar{1})\alpha_D$. This angle aligns well with data measured by optical microscopy for **1-Cl** (22°) and is comparable to **1-F** (21.23°) and **1-Br** (22.06°) (Fig. 2d). The consistent bending angle among **1-F**, **1-Cl**, and **1-Br** can be attributed to their isomorphous crystal packing. Interestingly, the crystal of

**1-Cl** displayed deformations with varying bending angles, such as 22° and 11°, where the twin interfaces are $(0\bar{1}1)\alpha_M//(01\bar{1})\alpha_D$ and $(0\bar{2}1)\alpha_M//(02\bar{1})\alpha_D$ respectively (Fig. 2d and Supplementary Fig. 12). These variable bending angles are attributed to multiple mechanical twinning modes about a zone axis.

Isomorphous crystal structures provide valuable insights into diverse interactions and distinct properties. While the isostructurality of Cl and CH₃ has been studied extensively[37], the isostructural nature of halogen groups (F/Cl/Br) has been relatively understudied[38]. In this study, **1-F**, **1-Cl**, and **1-Br** crystals served as a platform to investigate structural-level variations influencing deformation traits, such as coercive stress in ferroelastic deformable crystals (Figs. 1a and 2a). We also aimed to unveil the reasons behind variations in the coercive and reverse stress of these isomorphous crystals from a supramolecular perspective. In the crystal structure of **1-Cl**, molecules form dimeric synthons along the crystallographic c-axis through C–H⋯O hydrogen bonds (3.529 Å, 167.07°, and 3.3579 Å, 120.27°), while relatively weak C–H⋯Cl hydrogen bonds (3.09 Å, 3.127 Å, and 3.22 Å) connect these

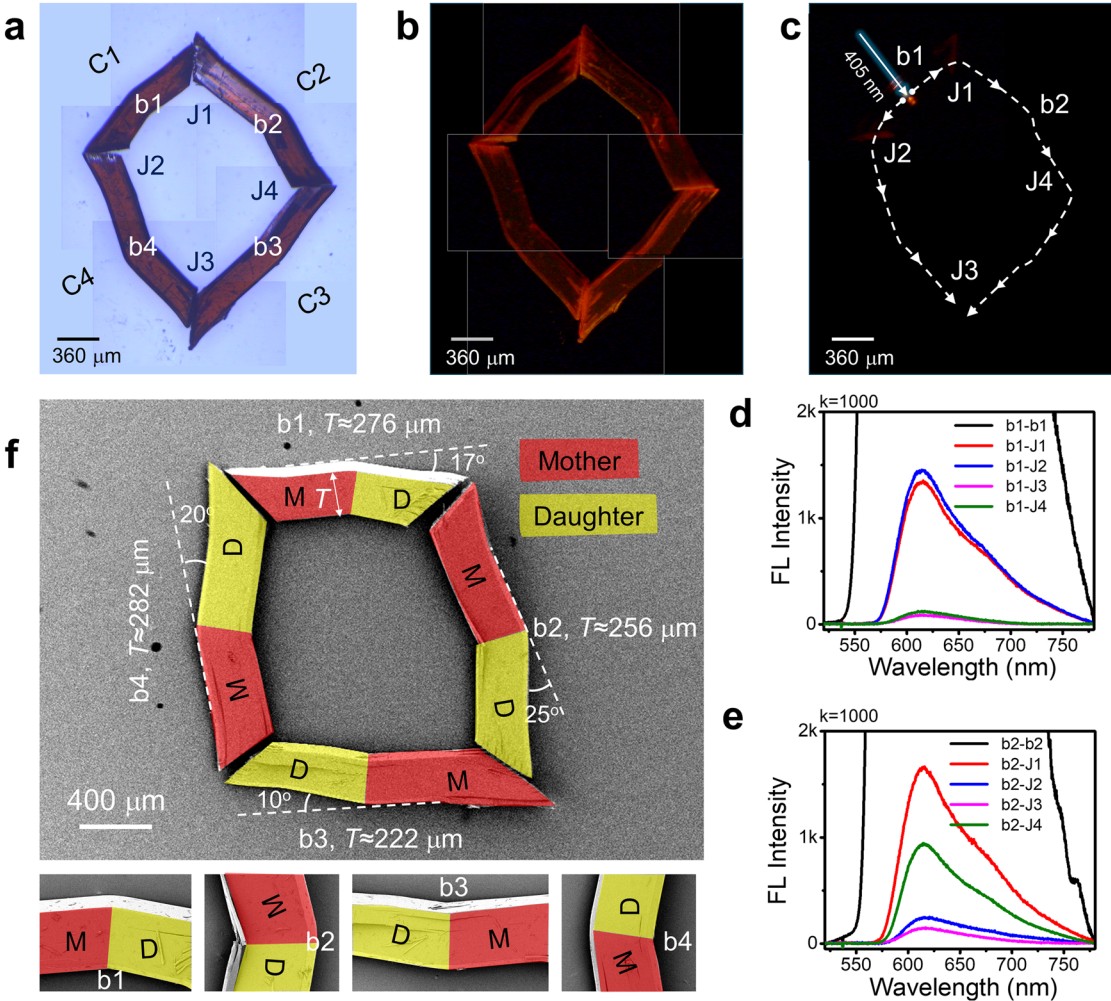

**Fig. 4 | Photonic aspects of the fabricated closed-loop structure from bent 1-Cl crystal waveguides. a** Stitched confocal optical and (**b**) FL microscope images (excited with UV torch) of closed-loop structure constructed from ferroelastic bent **1-Cl** crystal waveguides, presented on light blue and black colour background panels, respectively. C1-C4, b1-b4 and J1-J4 correspond to ferroelastic bent crystals, bends and junctions, respectively. **c** Stitched FL images of the closed-loop structure shown in a, when illuminated with 405 nm laser light at the bend, b1. The dotted white arrows show light propagation paths. **d, e** FL spectra recorded at different positions on the closed-loop structure when excited at b1 and b2, respectively. **f** Color-coded FESEM image of closed-loop structure shown in a. Bottom insets present the close-up view of the bent portions b1-b4, respectively. (Note: The position of crystals in FESEM was slightly altered while transferring onto the copper substrate to record electron microscope images).

dimers, creating 1D tapes along the crystallographic *a*-axis. The molecules exhibit π···π stacking interactions (shortest 3.402 Å, 3.430 Å) along the crystallographic *b*-axis, while the flexible side faces demonstrate a crisscross packing with a corrugation angle of approximately 85°. A similar interaction pattern is seen in the crystal structure of **1-Br**, with the Br···Br interaction being relatively stronger than the Br···Cl interaction (3.79 Å vs. 3.837 Å), corresponding to the order of coercive stress responsible for ferroelastic deformation.

Notably, the crystal structure of **1-F** exhibited significant differences in intermolecular interactions compared to **1-Cl** and **1-Br**, with stronger C−H···O and C−H···F interactions, along with prominent π···π stacking (3.369 Å). These distinct molecular interactions in **1-X** (**X** = **Cl**, **Br**, **I**) crystals likely contribute to variations in coercive stress. Specifically, the robust C−H···F interactions may account for the highest coercive stress observed, consistent with shear test measurements of **1-F** crystals, while stronger π···π interactions are responsible for the highest elastic modulus in the case of **1-F**.

Solid-state optical studies of the three isomorphous **1-X** crystals showed comparable optical properties, with their absorption spectra extending up to 620 nm (Supplementary Fig. 13 and Table 3). The

emission spectra covered the red part of the visible spectrum with slight variations in maxima (597 nm for **1-Br**, 607 nm for **1-F**, and 610 nm for **1-Cl**), and the absolute PL quantum yield was estimated to be less than 1% using an integrated sphere setup.

The **1-X** organoferroelectric crystals exhibited good strain-holding capacity, and their PL enabled the construction of sharply bent crystal waveguides, distinguishing them from elastic or plastic crystals. The **1-Cl** crystal (OW1, length, *L*≈1.35 mm; width, *W*≈276 μm) was excited with a 405 nm (continuous wave) diode laser at the left terminal, resulting in red fluorescence, FL (signal: λ(A) ≈552−780 nm; A denotes the active signal) with a λ$_{max}$ of ≈610 nm (Fig. 3a−c). The red FL propagated to the opposite right terminal through the crystal's long axis via the total internal reflection and outcoupled a reabsorbed signal -λ(A) ≈578−780 nm (-sign denotes the reabsorbed narrow band active signal), confirming the active type light transducing ability of OW1. The crystal was excited at various locations (1.35, 1.10, 0.90, 0.73, 0.53, 0.23, and 0 mm) starting from the right terminal, and their respective FL spectra were recorded (Fig. 3b). Consistent with previous studies[22,32–35], as the distance between the excitation and the detection points decreased, there was an exponential increase in FL intensity

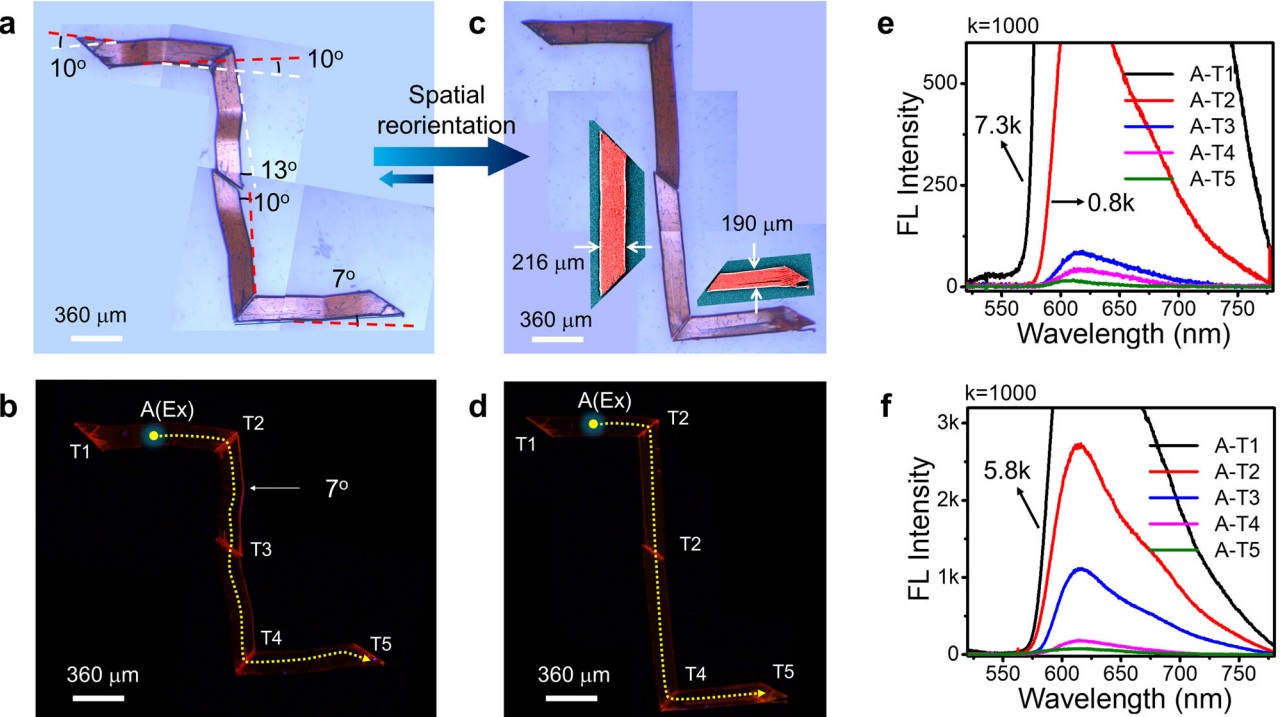

**Fig. 5 | Spatially controllable and geometrically reorientable optical wave-guiding through orthogonally arranged photonic structure. a, b** Stitched confocal optical and FL microscope images of Z-shaped photonic structure constructed from ferroelastic bent **1-Cl** crystal waveguides. **c, d** The stitched confocal optical and FL microscope images of reconfigured Z-shaped structure fabricated from spatially restored **1-Cl** crystal waveguides. Insets display the color-coded FESEM images of spatially restored **1-Cl** crystals. **e, f** The emission spectra recorded at various positions on the Z-shaped structure fabricated from bent and spatially restored **1-Cl** crystal waveguides, respectively. The blue arrows between (**a, c**) indicate the mechanical spatial reorientation possible in ferroelastic crystals. In (**a, c**), the images are presented on light blue colour background panels, for clarity. In (**b, d**), the images are presented on black background panels, for clarity.

(Fig. 3c). The optical loss (α′) for the straight OW1 was calculated to be ≈0.378 dB cm$^{-1}$ (Fig. 3g, Supplementary Fig. 14).

A critical understanding of how the optical properties of a material change under mechanical stress is essential for developing efficient organic photonic devices. To investigate the mechanophotonic aspects of the **1-Cl** crystal, OW1 was subjected to stress on the (00$\bar{1}$) plane. Initially, the crystal was deformed in the 'x' direction at 350 μm away from the left end, leading to a molecular rotation of 13° creating a daughter domain (as described in Supplementary Movie 1, Fig. 3d). Similar stress-induced ferroelastic deformations at 655 μm and 1 mm, with an angle of 13°, resulted in multiple bends in the **1-Cl** crystal. The FESEM images of the top- and tilted side-views of the bent crystal clearly illustrate the geometric changes (Fig. 3h, i). Optical waveguiding experiments conducted on the ferroelastic bent **1-Cl** crystal exhibited a waveguiding tendency as observed before the mechanical bending (Fig. 3e, f); however, the guided optical signal must navigate through the crystal's several sharply bent geometry (Fig. 3d). The α′ of the ferroelastic bent **1-Cl** crystal OWG was calculated to be 0.555 dB cm$^{-1}$ (Fig. 3g, Supplementary Fig. 14). The increased optical loss in the bent OWG is attributed to light scattering at the twin domain wall-crystal bend interface induced by mechanical bending and minor crystal damage during mechanical deformation. The light-guiding property on a high-aspect-ratio **1-Cl** crystal with more bends (eight bends) demonstrates impressive mechanophotonic characteristics (Supplementary Fig. 15a–c, g). Notably, the guided optical signal can sustain multiple ferroelastic deformations before outcoupling at the opposite terminal (Supplementary Fig. 15d–h). The increase in α′ after a series of bends on the straight crystal is only 0.425 dB cm$^{-1}$, validating the robustness of **1-Cl** ferroelastic crystal OWGs (Supplementary Fig. 15i).

Organic OWGs' suitability for photonic applications depends on their ability to construct diverse optical geometries, including closed-loop configurations. Achieving a closed-loop geometry involves integrating four bidirectional ferroelastic **1-Cl** crystals (C1-C4), each with mother and daughter domains. For that, a long crystal was cut with a razor blade into four fragments of comparable length at an approximate angle of 45° (Supplementary Movie 8), followed by deforming them ferroelastically (b1, b2, b3 and b4). The closed-loop structure was created by careful mechanical positioning of the bent crystal waveguides using tweezers under a confocal optical microscope (Fig. 4a, Supplementary Fig. 16). The FL image of the closed-loop structure showed intact waveguiding properties by exhibiting bright crystal termini. The color-coded FESEM image of the closed-loop structure revealed varying rotation angles (17°, 25°, 10°, and 20°) of the mother and daughter domains in the bent crystals (C1-C4), with thicknesses (T) ranging from 222 to 282 μm (Fig. 4f). The close-up view of the FESEM images indicated the smooth surface of these ferroelastic bent crystals. The photonic performance of the closed-loop structure of **1-Cl** under focused laser beam excitation at bend b1 in crystal C1 produced a red FL (signal: λ(A) ≈ 552 – 780 nm). This signal propagated towards C1's terminals, resulting in a narrow band active optical signal (-λ(A) ≈ 578 – 780 nm) at junctions J1 and J2 and passively coupled into neighboring crystals C2 and C4, respectively (Fig. 4c, d). This passive light in C2 and C4 traveled in clockwise and counterclockwise directions, towards their terminals as λ(P) ≈ 578 – 780 nm (here P denotes passive) signals at junctions J3 and J4, respectively. Similarly, when laser light was directed at bend b2 in crystal C2, the active FL traveled to J1 and J4 within the same crystal waveguide (Fig. 4e). Simultaneously, the same signal was passively coupled to adjacent crystals and detected at J3 and J2. This effective optical signal transmission between

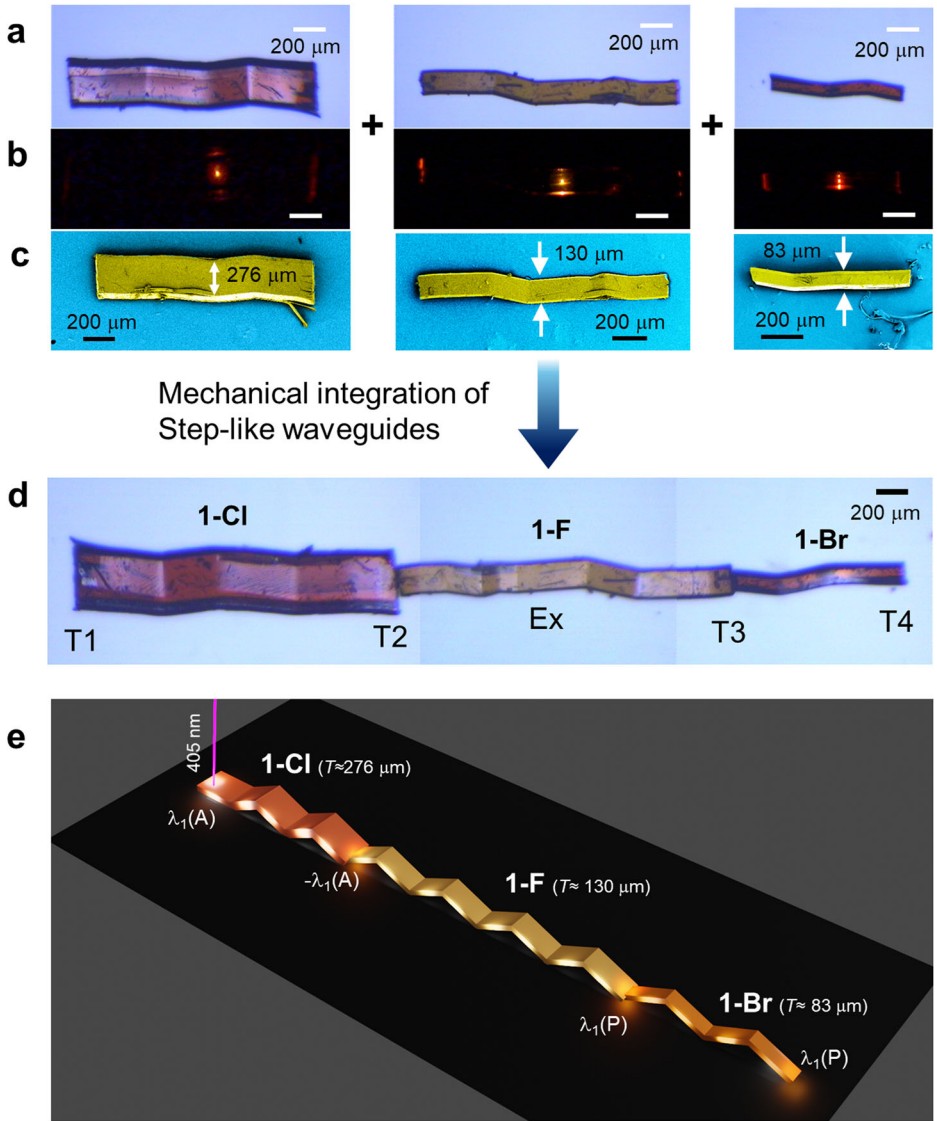

**Fig. 6 | Illustration of light propagation in a hybrid photonic waveguide.**
**a** Confocal optical, **b** FL and (**c**) corresponding color-coded FESEM, images of ferroelastic bent **1-Cl**, **1-F**, and, **1-Br** crystal waveguides, respectively. **d** Stitched confocal microscope images of a hybrid photonic structure fabricated by mechanical integration of crystals shown in (**a**). **e** Graphical illustration of light squeezing in the fabricated hybrid photonic structure. $T$ indicates the thickness of the crystal. T1-T4 stand for different crystal terminals. $\lambda_1(A)$ and $\lambda_1(P)$ correspond to active and passive signals operational in hybrid photonic waveguide.

bent crystals highlights the suitability of ferroelastic **1-Cl** crystal waveguides as monolithic interconnectors for photonic circuit applications.

To create spatially reconfigurable ferroelastic OWGs in two dimensions, four razor cut **1-Cl** crystal with a terminal angle of ≈45° were selected. These crystals were bent at angles of 7-13° and arranged into a Z-shaped photonic geometry with terminal-to-terminal contacts using a tweezer (Fig. 5a, b and Supplementary Fig. 17). The signal transmission along the Z-shaped structure was achieved by optically exciting the crystal at A (Ex), generating FL that guided to the crystal terminals, T1 and T2 within the same crystal. The light output at T2 coupled into adjacent crystal tips and propagated as a passive optical signal to T3, T4, and T5 using the terminal-to-terminal contacts (Fig. 5e). Notably, the optical signal efficiently transversed through two orthogonally positioned evanescently coupled bends in the Z-shaped structure. The α', as the light passes through multiple ferroelastic bends, could be reduced by reconfiguring the Z-shaped photonic structure by reorienting crystals to its straight geometry (Fig. 5c, d, f). This experiment demonstrates the feasibility of geometric adjustments in spatially controllable ferroelastic photonic structures.

Furthermore, to transduce the optical signal through a progressively narrow channel a hybrid photonic bent waveguide was created by interconnecting three different ferroelastically deformed crystal waveguides (**1-Cl**, **1-F**, **1-Br**) in decreasing order of thickness in a terminal-to-terminal coupled arrangement (Fig. 6a, b). The colour-coded FESEM images revealed that the chosen **1-Cl**, **1-F**, and **1-Br** crystals possess lengths of approximately 1.354, 1.510, and 0.731 mm, and thicknesses of about 276, 130 and 83 μm, respectively (Fig. 6c, Supplementary Figs. 18, 19). The mechanical integration of ferroelastically deformed crystals (**1-Cl**, **1-F**, and **1-Br**) in a terminal-to-terminal configuration created a 3.565 mm long hybrid photonic waveguide (Fig. 6d, e and Supplementary Figs. 20–22). The irradiation of a 405 nm laser beam on terminal T1 in **1-Cl** waveguide, resulted in FL (signal: λ(A) ≈ 552–780 nm) that propagated to T2, albeit with slight reabsorption (Fig. 6c, Supplementary Fig. 22b, c). The likelihood of energy transfer from **1-Cl** to **1-F** was minimal. Therefore, the same signal evanescently coupled into the adjacent **1-F** waveguide, reaching T3 as a passive signal

(-λ(P) ≈ 578–780 nm). Likewise, outcoupled light at T3 entered **1-Br** crystal waveguide, producing passive squeezed light output at T4 (Supplementary Fig. 22b, c). Shifting the input light to the middle of **1-F** crystal waveguide (labeled as "Ex" in Supplementary Fig. 22b) efficiently transfers the bright FL between termini T2 and T3, with passive signals outputs at T1 and T4, respectively (Supplementary Fig. 22c, d). For reverse light guiding from the thinner waveguide to the thicker ones, focusing the input light on the T4 terminal of **1-Br** waveguide resulted in optical signal propagation to T3, T2, and T1 across different waveguides (Supplementary Fig. 22a, b, e). The variation in the width of crystal waveguide termini and multiple bends resulted in coupling and bending-induced optical losses. Nevertheless, the staircase-like hybrid photonic waveguide geometry demonstrated squeezing optical signals across waveguides of different ferroelastic materials (Fig. 6e).

This study demonstrated three isomorphous organic crystals (**1-F**, **1-Cl**, and **1-Br**) with photoluminescent and versatile bi-directional ferroelastic properties. Their adaptable mechanical and photonic property enabled the fabrication of organic OWGs with sharp-bends. The bidirectional nature of ferroelastic crystals allowed us to create reconfigurable and sharply bent waveguides in closed-loop geometries, with control over optical signal's direction and intensity. Furthermore, integrating three chemically distinct ferroelastic crystals resulted in a staircase-like hybrid photonic waveguide with a light-squeezing capability. These findings uncover the potential of bidirectional ferroelastic organic crystals for diverse device applications, including smart optical sensors with structural reconfigurability.

## Data availability

The X-ray crystallographic coordinates for structures reported in this study have been deposited at the Cambridge Crystallographic Data Centre (CCDC), under deposition numbers CCDC 2308107-2308112. These data can be obtained free of charge from The Cambridge Crystallographic Data Centre via www.ccdc.cam.ac.uk/data_request/cif. The data that support the plots within this paper and other finding of this study are available within this article and its supplementary information file, and are also available from the corresponding author upon request.

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

## Acknowledgements

Funding was provided by MEXT KAKENHI (Grants Numbers JP22K18333 and JP22H02137), and JST CREST (Grant number JPMJCR23L2) for S.T. R.C. thanks SERB-NewDelhi (SERB-STR/2022/00011 and CRG/2023/003911) for financial support. S.R. thanks the Ministry of Education, Culture, Sports, Science and Technology (MEXT) for providing the MEXT fellowship.

## Author contributions

S.R. carried out the bulk crystal's bending and ferroelastic experiments, and analyzed the results under the supervision of S.T. A.V.K. conducted the photonics experiments and analyzed the results under the supervision of R.C. All authors contributed to the writing of the manuscript. S.R. and A.V.K equally contributed to this work.

## Competing interests

The authors declare no competing interest.
