## [Peer Review File · Nature Communications]

Spatially Controllable and Mechanically Switchable
Isomorphous Organoferroelectric Crystal Optical Waveguides
and NetworksREVIEWER COMMENTS

Reviewer #1 (Remarks to the Author):

The manuscript by Ranjan et al. reported versatile bi-directional ferroelasticity and optical waveguide properties of three isomorphous, halogen-based, Schiff base organic crystals. As summarized by the authors, the most outstanding attributes of these materials are their capabilities of sharp bending and dynamic shape reconfiguration. These findings are interesting to some extent. But they are not significant enough to be published in Nature Communications. There are also some obvious shortcomings.

1. The reviewer does not quite understand why the bending optical waveguide is needed. At current stage, it should be acknowledged that the future optical network or photonic circuits can not be built just by mechanical bending or micromanipulation. This is so inefficient and useless for real-world applications.
2. In Page 5, the authors stated “Moreover, surface roughness measurements by laser confocal microscopy revealed that...” To my point of view, the surface roughness should not be examined by laser confocal microscopy. AFM can provide more direct and accurate evidence into surface roughness.
3. The absolute PLQY of all three materials investigated in this work is below 1%, which is too low to be considered for practical applications as proposed in this work.
4. In Figure 4, a closed loop structure was constructed. But the reviewer is not convinced why the bended waveguide is used here. In other words, a straight waveguide can also fulfill this purpose with better performance.

Overall, the reviewer feels that the results presented this work can not support most of this claim for photonic network or applications. Therefore, I can not recommend the publication of this work.

Reviewer #2 (Remarks to the Author):

Satoshi Takamizawa and coauthors designed and synthesized three new halogen-based Schiff base organic crystals, which showed versatile bi-directional ferroelasticity and optical waveguide properties. The data and discussion in this paper are well understandable to give the conclusion with novel results. The manuscript was recommended for publication after careful revisions. Some suggestions are listed:

1. The authors are suggested to carefully polish the English language and check the grammar, format. Please double-check the whole paper as well as the citation of the References.
2. There are some mistakes in Table S1, “aM (1-Cl)” in the last penultimate column should be “aM (1-Br)” and “aD (1-Cl)” in the last column should be “aD (1-Br)”.
3. The DFT calculations should be provided to support their absorption and emission spectra.

4. Optical wave-guiding characteristics of 1-F and 1-Br crystal should be provided.
5. Please further explain why these crystals gave lower optical loss.

Reviewer #3 (Remarks to the Author):

The article by Subham Ranjan et al., presents a comprehensive study on the synthesis, characterization, and application of three isomorphous Schiff base organic crystals (designated as 1-X; X = F, Cl, Br) with ferroelastic and photoluminescent properties for use in organic optical waveguides (OWGs). Through meticulous experimental procedures, the authors have demonstrated the crystals' capability for versatile mechanical deformation while preserving their photonic properties, a significant advancement for programmable photonic devices. This work represents a significant contribution to the fields of materials science and photonics, particularly in the development of flexible, reconfigurable optical waveguides. Compared to existing materials, these organoferroelastic crystals offer a unique combination of mechanical flexibility and optical functionality, positioning them as promising candidates for advanced optical devices, sensors, and circuits. While organic ferroelastic materials have been previously studied, the specific combination of ferroelasticity, photoluminescence, and waveguiding capabilities in a single material system, as demonstrated here, is relatively novel. Prior works have focused on either aspect separately, but the integration into organoferroelastic crystals for photonic applications marks a notable advancement. For instance, the use of photoluminescent ferroelastic crystals as optical waveguides (OWGs) represents a merger of mechanical and optical sciences that has not been extensively explored in the current literature. The authors have thoroughly characterized the mechanical and optical properties of the crystals, including their ferroelastic deformation, photoluminescence, and waveguiding capabilities. The use of single-crystal X-ray diffraction, FESEM imaging, and stress-strain analysis provides a solid foundation for the observed phenomena. The ability to construct Z-shaped, closed-loop, and staircase-like photonic structures with controlled light path manipulation showcases the potential for integrating these materials into next-generation optical devices, sensors, and circuits. While it looks like an interesting article. Some minor modifications and clarifications could make it potentially appealing to a wider readership

1. The authors reported a quantum yield of less than 1% for the synthesized materials. Given the relatively low efficiency in converting absorbed light into emitted light, could you elaborate on other applications (besides waveguides) of these materials where such a quantum yield would be considered adequate? Specifically, how does this quantum efficiency impact the performance in applications not primarily focused on light emission, such as waveguides or sensor components? Furthermore, in applications requiring active light emission (e.g., OLEDs, laser sources, or high-sensitivity fluorescence-based sensors), a higher quantum yield could significantly enhance the material's utility. Are there ongoing efforts or future plans to improve the quantum yield through molecular engineering, incorporation into composite materials or structures, or synergistic

combinations with other high-quantum-yield materials to boost overall device performance? Your insights on these aspects would greatly contribute to understanding the full potential and scope of applications for these innovative materials.

2. While the article discusses the mechanical deformation and elastic moduli of the crystals, there is less emphasis on long-term mechanical stability and durability under repeated deformation, which are crucial for practical applications. Any comments would be welcomed.

3. The synthesis and characterization processes are well-detailed for laboratory-scale experiments, Can the authors address the scalability of these materials for industrial application, including potential challenges in large-scale manufacturing and integration into devices. This would be beneficial to the wider audience readership of Nature Communications.

RESPONSES TO REVIEWERS

We would like to sincerely thank all of you for the careful evaluation and helpful comments on our manuscript. Our responses and revisions are below.

Reviewer(s)' Comments to Author:

Reviewer: 1

Recommendation: Findings are interesting to some extent but not significant enough to be published in Nature Communications.

The manuscript by Ranjan et al. reported versatile bi-directional ferroelasticity and optical waveguide properties of three isomorphous, halogen-based, Schiff base organic crystals. As summarized by the authors, the most outstanding attributes of these materials are their capabilities of sharp bending and dynamic shape reconfiguration. These findings are interesting to some extent. But they are not significant enough to be published in Nature Communications. There are also some obvious shortcomings.

Response: We thank the reviewer for accepting that the results are indeed interesting to the scientific community. Please find our responses for a detailed understanding.

Comments:

1. The reviewer does not quite understand why the bending optical waveguide is needed. At current stage, it should be acknowledged that the future optical network or photonic circuits can not be built just by mechanical bending or micromanipulation. This is so inefficient and useless for real-world applications.

Response:

Thank you for your comments. To answer this, it is important to highlight the prospective applications of flexible organic crystals as next-generation flexible electronic and optoelectronic materials which have been strongly acknowledged by some of the recent review articles.¹⁻³ Flexible organic crystals are potential candidates for next-generation flexible electronics such as wearable and smart sensors, displays, artificial skins, and soft robotics. Flexible organic crystals

offer great advantages in terms of flexibility, molecular diversity, easy synthesis, and less grain boundaries. In contrast to organic electronics, photonic organic crystals with high refractive index enable faster information transfer with minimal optical loss. The widely used silicon-based photonic circuits provide better light confinement with low optical loss but are rigid and inflexible. As a result, flexible organic crystals have the potential to be used in next-generation flexible optoelectronics and have been demonstrated to show optical waveguiding properties in optical micro/nanocircuits.^{4,5} Presently, flexible LED displays, wearable health sensors, and foldable cell phones are already in the marketplace. Researchers have successfully integrated flexible organic semiconductors and photonic organic crystals as prototype elements into circuits, thereby providing unequivocal proof of their potential applications in flexible optoelectronics microcircuits.^{4,5}

In this study, we reported the first optical waveguiding properties in ferroelastic organic crystals. So far, the optical waveguiding properties of organic single crystals with elastic/plastic properties have been investigated for both active and passive light transduction. However, crystalline integrity in the deformed region and sharp bends cannot be achieved with elastic/plastic crystals. On the other hand, the diffusionless twinned domain of ferroelastically bent crystals can eliminate this limitation and enable better control and guidance of light waves at sharp bends which are important for efficient light transduction. Furthermore, the millimeter-sized ferroelastic crystals (this study) can be easily bent by a mechanical force of tweezers and do not require micromanipulation.

Therefore, analyzing the prospective applications and results from the published papers and our work, it is unlikely to rule out the utility of mechanically bendable crystals for photonic circuit applications. For instance, during the early 1980s, researchers thought organic materials would not be suitable for LED applications (due to low stability and difficulty in device integration). However, researchers have found different ways to improve the performance of organic materials by modulating their emission color, contrast, efficiency, and more. Therefore, these initial results about sharply bendable waveguides may open up new possibilities in reconfigurable photonic circuits, which form the fundamental basis for programmable photonic circuits and other device technologies.

For a better understanding of the potentiality of our present work, we have included some additional lines in the introduction. We have also added millimeter-order crystals in the manuscript.

References

- (1) Wang, Y.; Sun, L.; Wang, C.; Yang, F.; Ren, X.; Zhang, X.; Dong, H.; Hu, W. Organic crystalline materials in flexible electronics. *Chem. Soc. Rev.* **2019**, *48*, 1492-1530.
- (2) Jiang, H.; Hu, W. The emergence of organic single-crystal electronics. *Angew. Chem. Int. Ed.* **2020**, *59*, 1408-1428.
- (3) Wei, C.; Li, L.; Zheng, Y.; Wang, L.; Ma, J.; Xu, M.; Lin, J.; Xie, L.; Naumov, P.; Ding, X. Flexible molecular crystals for optoelectronic applications. *Chem. Soc. Rev.* **2024**.
- (4) Karothu, D. P.; Dushaq, G.; Ahmed, E.; Catalano, L.; Polavaram, S.; Ferreira, R.; Li, L.; Mohamed, S.; Rasras, M.; Naumov, P. Mechanically robust amino acid crystals as fiber-optic transducers and wide bandpass filters for optical communication in the near-infrared. *Nat. Commun.* **2021**, *12*, 1326.
- (5) Ravi, J.; Feiler, T.; Mondal, A.; Michalchuk, A. A.; Reddy, C. M.; Bhattacharya, B.; Emmerling, F.; Chandrasekar, R. Plastically Bendable Organic Crystals for Monolithic and Hybrid Micro-Optical Circuits. *Adv. Opt. Mater.* **2023**, *11*, 2201518.

2. In Page 5, the authors stated “Moreover, surface roughness measurements by laser confocal microscopy revealed that...” To my point of view, the surface roughness should not be examined by laser confocal microscopy. AFM can provide more direct and accurate evidence into surface roughness.

Response: Thank you for highlighting this point. Certainly, AFM can provide conclusive proof of surface roughness, but domain contrast will be a challenge to directly visualize the distinct domains. The 3D laser confocal scanning microscopy provides fast image acquisition, a wider area with high-resolution microscope images, and precise submicron distance measuring along the XY axes. Considering its excellent crystal-clear 3D images and its capacity to produce high-resolution and high-contrast 3D imagery without background interference, we used it to examine the surface roughness with a contrast imaging of mother and daughter domain where we can be easily perceived and distinguish the mother and daughter domains with its roughness in updated figure S7.

3. The absolute PLQY of all three materials investigated in this work is below 1%, which is too low to be considered for practical applications as proposed in this work.

Response: Thank you for the query. The utility of low PLQY materials for different applications can be perceived depending on the intended use. Here, these low PLQY materials can be put to

good use for passive waveguiding applications, as illustrated in the hybrid optical waveguide (Figure 6, in the manuscript).

4. In Figure 4, a closed loop structure was constructed. But the reviewer is not convinced why the bended waveguide is used here. In other words, a straight waveguide can also fulfill this purpose with better performance.

Response: Thank you for your constructive comment. In Figure 4, a closed loop structure was constructed.

We agree that straight waveguides can also be used to construct closed-loop structures, but the junctions created in the case of straight waveguides possess right-angle contact points. Directing light in orthogonal directions efficiently is always a challenge in photonics. However, the use of bent waveguides reduces the contact angle below 90° and ensures effective light coupling from one waveguide to another.

Furthermore, the primary aim of using bent crystals as an optical waveguide is to integrate them into flexible optoelectronic microcircuits such as wearable sensors. As already mentioned, integrating straight organic crystals to form a closed-loop structure will only result in a 90° sharp bend. However, a closed loop structure with various desired bending angles can be achieved by versatile organoferroelastic crystals, which has been demonstrated in our present study. For flexible wearable sensors to be affirmed in the hand or knee might need smaller bending angles than 90° for better mechanical compliance with skin.

Overall, the reviewer feels that the results presented this work can not support most of this claim for photonic network or applications. Therefore, I can not recommend the publication of this work.

Response: We hope that our response will convince you of the importance of ferroelastic crystal waveguides in a variety of optical-based device applications.

Reviewer: 2

Recommendation: The manuscript was recommended for publication after careful revisions.

Satoshi Takamizawa and coauthors designed and synthesized three new halogen-based Schiff base organic crystals, which showed versatile bi-directional ferroelasticity and optical waveguide properties. The data and discussion in this paper are well understandable to give the conclusion with novel results. The manuscript was recommended for publication after careful revisions. Some suggestions are listed:

Response: We are grateful to Reviewer 2 for giving helpful comments on our manuscript.

1. The authors are suggested to carefully polish the English language and check the grammar, format. Please double-check the whole paper as well as the citation of the References.

Response: Thanks for the suggestion. The manuscript has been revised, and necessary corrections have been incorporated wherever necessary.

2. There are some mistakes in Table S1, “ α_M (1-Cl)” in the last penultimate column should be “ α_M (1-Br)” and “ α_D (1-Cl)” in the last column should be “ α_D (1-Br)”.

Response: Thank you for pointing out this typographical error. We have corrected Table S1 in SI. The corrected text is as follows: α_M (1-Br) and α_D (1-Br).

3. The DFT calculations should be provided to support their absorption and emission spectra.

Response: Thank you for for this query. We have done additional experiments on the solution state optical absorption and emission studies and performed DFT calculations which are incorporated in the supplementary information 13.

4. Optical wave-guiding characteristics of 1-F and 1-Br crystal should be provided.

Response: Thank you for the query. We have previously provided the optical waveguiding characteristics of 1-F and 1-Br in the supporting information in Supplementary figures S16 and S17, respectively. Please see the below figures.

Figure S16. Confocal optical and FL images of **a,b** straight and **c,d** ferroelastic bent **1-Br** crystal waveguide, respectively. **e,f** Excitation position-dependent emission studies performed on a straight and ferroelastic bent **1-Br** crystal waveguide, respectively. **g** Optical loss observed before and after ferroelastic bending in **1-Br** crystal waveguide.

Figure S17. Confocal optical and FL images of **a,b** straight and **c,d** ferroelastic bent **1-F** crystal waveguide, respectively. Note. The crystal in **a** and **c** are viewed along thinner and thicker facets, respectively. **e** Excitation position-dependent emission spectra recorded on a **1-F** ferroelastic bent crystal waveguide. **f** Optical loss calculated for the **1-F** ferroelastic bent crystal waveguide.

5. Please further explain why these crystals gave lower optical loss.

Response: Thank you for the query. The optical loss of a waveguide depends on several parameters. However, in an active-type waveguide, optical loss is primarily governed by the quality of the crystal, surface scattering losses, and reabsorption losses. Here, the high-quality defect-free crystals with smooth surface morphology reduce the light loss. Further, a smaller reabsorption region also contributes to the high efficiency of the waveguides. Therefore, the optical losses were observed to be minimal in these crystal waveguides.

Reviewer: 3

Recommendation: Some minor modifications and clarifications could make it potentially appealing to a wider readership.

The article by Subham Ranjan et al., presents a comprehensive study on the synthesis, characterization, and application of three isomorphous Schiff base organic crystals (designated as 1-X; X = F, Cl, Br) with ferroelastic and photoluminescent properties for use in organic optical waveguides (OWGs). Through meticulous experimental procedures, the authors have demonstrated the crystals' capability for versatile mechanical deformation while preserving their photonic properties, a significant advancement for programmable photonic devices. This work represents a significant contribution to the fields of materials science and photonics, particularly in the development of flexible, reconfigurable optical waveguides. Compared to existing materials, these organoferroelastic crystals offer a unique combination of mechanical flexibility and optical functionality, positioning them as promising candidates for advanced optical devices, sensors, and circuits. While organic ferroelastic materials have been previously studied, the specific combination of ferroelasticity, photoluminescence, and waveguiding capabilities in a single material system, as demonstrated here, is relatively novel. Prior works have focused on either aspect separately, but the integration into organoferroelastic crystals for photonic applications marks a notable advancement. For instance, the use of photoluminescent ferroelastic crystals as optical waveguides (OWGs) represents a merger of mechanical and optical sciences that has not been extensively explored in the current literature. The authors have thoroughly characterized the mechanical and optical properties of the crystals, including their ferroelastic deformation, photoluminescence, and waveguiding capabilities. The use of single-crystal X-ray diffraction, FESEM imaging, and stress-strain analysis provides a solid foundation for the observed phenomena. The ability to construct Z-shaped, closed-loop, and staircase-like photonic structures with controlled light path manipulation showcases the potential for integrating these materials into next-generation optical devices, sensors, and circuits. While it looks like an interesting article. Some minor modifications and clarifications could make it potentially appealing to a wider readership.

Response: We thank the reviewer for the thorough analysis of the present work and his appreciative words.

1. The authors reported a quantum yield of less than 1% for the synthesized materials. Given the relatively low efficiency in converting absorbed light into emitted light, could you elaborate on

other applications (besides waveguides) of these materials where such a quantum yield would be considered adequate? Specifically, how does this quantum efficiency impact the performance in applications not primarily focused on light emission, such as waveguides or sensor components?

Response: Thanks for raising such an important point. The quantum yield of a material plays an important role while working exclusively with active-type optical waveguides. However, the spontaneous use of different materials (multiple crystals with distinct emission properties) for photonic circuit applications provides a facile route to utilize materials with varying emission properties.

The utility of low PLQY materials for different applications can be perceived depending on the intended use. Here, these low PLQY materials can be put to good use for passive waveguiding applications, as illustrated in the hybrid optical waveguide (Figure 6, in the manuscript). Materials with high PLQY are useful for active waveguiding and coupling components such as another waveguides and resonators.

Furthermore, in applications requiring active light emission (e.g., OLEDs, laser sources, or high-sensitivity fluorescence-based sensors), a higher quantum yield could significantly enhance the material's utility. Are there ongoing efforts or future plans to improve the quantum yield through molecular engineering, incorporation into composite materials or structures, or synergistic combinations with other high-quantum-yield materials to boost overall device performance? Your insights on these aspects would greatly contribute to understanding the full potential and scope of applications for these innovative materials.

Response: Thanks for the point. We completely agree with you. The use of materials in OLEDs, and related applications strictly require high quantum efficiency. Studies are underway in our laboratories to construct high PLQY crystals through molecular engineering principles.

2. While the article discusses the mechanical deformation and elastic moduli of the crystals, there is less emphasis on long-term mechanical stability and durability under repeated deformation, which are crucial for practical applications. Any comments would be welcomed.

Response: Thank you for your comment. Keeping the stability in mind, we have tested the ferroelastic deformation in freshly obtained crystals and 13-month-old crystals. No difference was observed in the deformation behavior or crystallinity in the crystals (Fig. A).

Forward transformation

Reverse transformation

Fig A. Ferroelastic deformation in 13 months old crystals of **1-Cl**.

3. The synthesis and characterization processes are well-detailed for laboratory-scale experiments, Can the authors address the scalability of these materials for industrial application, including potential challenges in large-scale manufacturing and integration into devices. This would be beneficial to the wider audience readership of Nature Communications.

Response: Organic thin films are generally used in commercial electronic devices rather than single crystals. However, investigating single crystals for establishing photonic or electronic applications allows an easy understanding of the structure-property relationship and the causative mechanism. At this stage, we are focused on establishing a guide for developing such organoferroelastic crystals and finding design principles. For futuristic applications, large-size organic crystals can be obtained by epitaxial growth and organic thin films can be made by spin coating, or spraying methods, or printing technologies. Furthermore, organic single crystals possess inherent compatibility with flexible substrates which can be used for integration into devices.

REVIEWER COMMENTS

Reviewer #1 (Remarks to the Author):

I would like to thank the authors for their efforts to address my concerns. Unfortunately, I am not fully convinced by their responses to the significance of this work and the interpretation to some experimental results.

In terms of significance, the authors are arguing that flexible organic crystals are promising candidates for future photonic circuits by referring to flexible organic electronics. But there are so many issues need to be addressed before introducing it to photonics. Then, the authors mentioned the development of LED has the similar issues. Realistically, the first LED paper was published on Applied Physics Letters. In addition, the most critical issue is that the connection between each individual component can be easily broken down during the bending of real device, which makes it almost impossible for practical applications as proposed by the authors. Therefore, I do not think developing a bendable optical waveguide can attract broad interest at current stage.

In terms of roughness, fine, you can retrieve this information from laser confocal microscope. But the roughness value of 57 nm is too large for a single crystal. Typically, a nice single crystal can show a roughness down to 1 nm. The single crystal with such a rough surface will induce significant optical loss due to surface scattering, which is not ideal for photonic applications. The authors should comment on this. Besides that, I still recommend AFM measurement to study surface roughness.

In terms of PLQY issue, I noticed that the reviewer #3 had the same concerns with me. The authors argued that this might be helpful for passive waveguide applications. But why did the authors measure and report the optical loss of active waveguiding? Why not directly measure the loss coefficient of passive waveguide? The active waveguide is radically different from passive waveguide. In the sense, it seems that the authors also do not know which direction this material should be put into use.

Still, the reviewer cannot feel the urgent need to publish this work in such a prestigious journal like Nature Communications. I will let the editor to make the final decision.

Reviewer #2 (Remarks to the Author):

the authors revised the manuscript well according to the suggestions of reviewers, the reviewer suggest the revised manuscript can be accepted.

Reviewer #3 (Remarks to the Author):

The authors have addressed my concerns and remarks.

Reviewer #1 (Remarks to the Author):

I would like to thank the authors for their efforts to address my concerns. Unfortunately, I am not fully convinced by their responses to the significance of this work and the interpretation to some experimental results.

(Res.) We thank the reviewer for his insights and comments on our work. However, we feel sorry that the reviewer does not appreciate the importance of the present work.

In terms of significance, the authors are arguing that flexible organic crystals are promising candidates for future photonic circuits by referring to flexible organic electronics.

But there are so many issues need to be addressed before introducing it to photonics. Then, the authors mentioned the development of LED has the similar issues. Realistically, the first LED paper was published on Applied Physics Letters. In addition, the most critical issue is that the connection between each individual component can be easily broken down during the bending of real device, which makes it almost impossible for practical applications as proposed by the authors. Therefore, I do not think developing a bendable optical waveguide can attract broad interest at current stage.

(Res.)

P
A
G
E

This paper demonstrates the usefulness of strong ferroelastic deformation as a new mechanism that can solve the previously thought impossible problem of real devices being easily destroyed during bending, even in organic single crystals, which are considered the most difficult. The ferroelastic deformation allows extremely sharp bends in the organic crystal waveguides and will be useful to route optical signals at sharp turns. These ferroelastically deformed optical waveguides were capable of transducing active/passive optical signals even when the crystals possessed up to eight sharp bends. Therefore, these bendable optical waveguides hold great significance for the advancement of flexible organic crystal-based photonic circuit technology and will attract huge attention from researchers across different areas.

In terms of roughness, fine, you can retrieve this information from laser confocal microscope. But the roughness value of 57 nm is too large for a single crystal. Typically, a nice single crystal can show a roughness down to 1 nm. A single crystal with such a rough surface will induce significant optical loss due to surface scattering, which is not ideal for photonic applications. The authors should comment on this. Besides that, I still recommend AFM measurement to study surface roughness.

(Res.) The crystallinity of bulk single crystals is the first important material requirement, and

P
A
G
E

the high crystallinity of single crystals is evident from single-crystal X-ray diffraction. Although there is not necessarily a correlation between crystal surface and single crystallinity, in the solution concentrated crystal growth method by solvent evaporation used here, the roughness value of the crystal surface by laser microscopy (which allows a relatively wide range of non-contact observation compared to AFM) is sufficiently acceptable to be considered a single-crystal surface. In our experience, we consider this sample to be of sufficiently high quality as an organic single crystal. The 1nm expectation presented by the reviewer (Typically, a nice single crystal can show a roughness down to 1 nm.) sounds like an misleading.

Further, as suggested by the reviewer, the AFM surface roughness measurements on a millimeter-sized ferroelastically bent crystal suggested an average roughness value between 13-49 nm, with a root mean square roughness of about 4-17, depending on the area of measurement.

Supplementary Fig. 23 has been added in ESI:

Supplementary Figure 23. a Snapshot of unevenly ferroelastically deformed crystal (1-CI), b Analysed area for surface roughness, c-g Surface topography from area #1 to #5, respectively. and h surface and root mean square roughness of area #1-#5.

The surface roughness values obtained for the unevenly bent crystal are lower than those reported for other optical waveguides.⁷

7. Sun, Shang, H., & Hiang, H. Effective metrology and standard of the surface roughness of micro/nanoscale waveguides with confocal laser scanning microscopy. *Opt. Lett.* **44**, 747-750 (2019).

In terms of PLQY issue, I noticed that the reviewer #3 had the same concerns with me. The authors argued that this might be helpful for passive waveguide applications. But why did the authors measure and report the optical loss of active waveguiding? Why not directly measure the loss coefficient of passive waveguide? The active waveguide is radically different from passive waveguide. In the sense, it seems that the authors also do not know which direction this material should be put into use.

(Res.) Thanks for the query. One of the major drawbacks of silicon photonics is that Si waveguides are majorly passive-only signal transducers. However, the advantage of organic waveguides is that they can act as active and/or passive waveguides depending on the input signal's wavelength (whether it falls in/out of the crystals' optical absorption region). Here, first, we demonstrate the active waveguiding capabilities of pristine crystals (1-X), followed by the same in ferroelastically bent crystals. We also depicted the passive waveguiding capabilities of ferroelastically bent crystals in the closed loop structures, reconfigurable Z-shaped photonic structure, and hybrid photonic waveguide. Hence, these ferroelastically bendable crystal waveguides can be used as active/passive waveguides, depending on their intended use. This trait, in combination with sharply bendable characteristics, makes ferroelastically bendable optical waveguides useful for photonic circuit applications.

Still, the reviewer cannot feel the urgent need to publish this work in such a prestigious journal like Nature Communications. I will let the editor to make the final decision.

(Res.) Unfortunately, the reviewer does not perceive the importance of this work. This joint collaborative research work explores the hitherto unknown sharply bendable ferroelastic organic optical waveguides from three isomorphous organoferroelastic crystals. The isomorphous nature of the crystals was studied using single-crystal X-ray diffraction and XPac

P
A
G
E

studies. For the first time, this work reports optical waveguiding in ferroelastic crystals. Moreover, photonic compatibility in terms of active/passive waveguiding in individual and hybrid photonic structures cements their capabilities for photonic circuit applications. Therefore, after reading our lines, we await the editor's sensible decision.

Reviewer #2 (Remarks to the Author):

the authors revised the manuscript well according to the suggestions of reviewers, the reviewer suggest the revised manuscript can be accepted.

(Res.) We thank the reviewer for his/her time and for recommending the publication of the paper.

Reviewer #3 (Remarks to the Author):

The authors have addressed my concerns and remarks.

(Res.) We thank the reviewer for his/her time and for recommending the publication of the paper.

P
A
G
E

Additional Changes:

We have added a new **Movie_S8** to help readers understand cutting crystals with a razor blade.

REVIEWERS' COMMENTS

Reviewer #1 (Remarks to the Author):

The revised version of the manuscript is now ready for publication.